# Association of Broad-Spectrum Antibiotic Therapy and Vitamin E Supplementation with Vitamin K Deficiency-Induced Coagulopathy: A Case Report and Narrative Review of the Literature

**DOI:** 10.3390/jpm13091349

**Published:** 2023-08-31

**Authors:** Andreas M. Matthaiou, Ioannis Tomos, Sofia Chaniotaki, Dimitrios Liakopoulos, Katerina Sakellaropoulou, Sofia Koukidou, Loredana-Mariana Gheorghe, Stefanos Eskioglou, Angeliki Paspalli, Georgios Hillas, Katerina Dimakou

**Affiliations:** 15th Department of Respiratory Medicine, Sotiria Thoracic Diseases General Hospital of Athens, 11527 Athens, Greece; 2Laboratory of Molecular and Cellular Pneumonology, Medical School, University of Crete, 71003 Heraklion, Greece; 3Respiratory Physiology Laboratory, Medical School, University of Cyprus, 2029 Nicosia, Cyprus; 4Department of Laboratory Haematology, Georgios Gennimatas General Hospital of Athens, 11527 Athens, Greece

**Keywords:** vitamin K, antibiotics, vitamin E, international normalised ratio, prothrombin time, coagulopathy

## Abstract

Vitamin K is a lipid-soluble vitamin that is normally maintained within appropriate levels by means of dietary intake and bacterial production in the intestinal microflora. It holds a central role in coagulation homeostasis, and thus its depletion leads to hypocoagulation and haemorrhagic diathesis. The association of antibiotic therapy and vitamin E supplementation with vitamin K deficiency was previously described in animal experiments, clinical studies, and case reports. Broad-spectrum antibiotic therapy potentially leads to intestinal microflora dysbiosis and restriction of vitamin K-producing bacterial populations, resulting in decreased vitamin K levels, whereas antibiotics of the cephalosporin class with 1-*N*-methyl-5-thiotetrazole (NMTT) or 2-methyl-1,3,4-thiadiazole (MTD) side groups inhibit vitamin K function. Vitamin E supplementation interferes with both the bioavailability and function of vitamin K, yet its mechanisms are not fully understood. We present the case of a 45-year-old male patient, with a history of epilepsy and schizophrenia, catatonically incapacitated and immobilised, who was hospitalised in our centre for the investigation and management of aspiration pneumonia. He demonstrated a progressively worsening prolongation of international normalised ratio (INR), which was attributed to both broad-spectrum antibiotic therapy and vitamin E supplementation and was reversed upon administration of vitamin K. We highlight the need for close monitoring of coagulation parameters in patients receiving broad-spectrum antibiotic therapy, especially those with underlying malnutritive or malabsorptive conditions, and we further recommend the avoidance of NMTT- or MTD-containing antibiotics or vitamin E supplementation, unless absolutely necessary, in those patients.

## 1. Clinical Vignette

A 45-year-old male of Greek origin was referred to the Emergency Department of the Sotiria Thoracic Diseases General Hospital of Athens due to fever, dyspnea at rest, and productive cough with purulent sputum within the last 24 h. The patient had a previous history of epilepsy and schizophrenia, being catatonically incapacitated, immobilised, and malnourished, while he had also presented multiple aspiration pneumonia episodes in the past. At presentation, clinical examination was remarkable for cachexia, while thoracic auscultation revealed diffuse rhonchi in all lung fields and prolongation of expiration. Laboratory examinations revealed high levels of inflammatory markers, including white blood cell count and C-reactive protein, and normal coagulation studies, including international normalised ratio (INR), prothrombin time (PT), and activated partial thromboplastin time (APTT), as shown in Table 1. Serial chest X-rays throughout hospitalisation showed opacities in the right lower lung field, as shown in Figure 1.

The patient was initially set on intravenous empirical antibiotic therapy with piperacillin/tazobactam, which was later switched to meropenem and vancomycin on the third day of hospitalisation due to persistent fever and increasing inflammatory markers, including procalcitonin (Table 1). He was also treated with subcutaneous enoxaparin in prophylactic dose, intravenous omeprazole, intravenous multivitamin supplementation (vitamins A, D_3_, E, C, B_1–3_, B_5–7_, B_9_, and B_12_), as well as his antiepileptic therapeutic regimen with clobazam, phenytoin, and valproate. Importantly, the patient did not receive any other anticoagulants except for enoxaparin, including any vitamin K antagonists.

During his hospitalisation, the patient demonstrated extended opacities in the right lung fields, as shown in a repeat chest X-ray, as well as worsening respiratory failure. Chest computed tomography revealed complete atelectasis of the right lung due to occlusion of the right main bronchus by impacted airway secretions and a small ipsilateral pleural effusion. Bedside bronchoscopy was performed to remove the endobronchial secretions, and the right lung eventually re-expanded, as shown in serial chest X-rays. On the 13th day of hospitalisation, the patient manifested relapsing fever and increasing inflammatory markers, while *Acinetobacter baumannii* was isolated from the bronchial washing culture. Meropenem and vancomycin were thus discontinued, and targeted antibiotic therapy with colistin, tigecycline, and ampicillin/sulbactam was started based on the antibiogram.

Serial coagulation studies revealed a progressively worsening prolongation of INR and PT, with only mild prolongation of APTT, after the first week of hospitalisation, despite the discontinuation of enoxaparin, without any deterioration of liver function tests. On the 19th day of hospitalisation, INR demonstrated a further severe increment reaching 7.94, with PT: 91.6 s and APTT: 66.2 s. The patient did not present any clinical signs of haemorrhage or any significant drop in the haematocrit and haemoglobin levels. He was immediately intravenously administered with 10 mg of vitamin K_1_ and then a repeat assessment of coagulation status after 6 h revealed a rapid fall of INR to 2.86, with PT: 35.2 s and APTT: 69.3 s. Additionally, a mixing test of the patient’s blood with control blood was performed and showed a further restoration of INR to 1.23. Factor VII activity was found to be extremely low at 8.40% (reference values: 50–130%). After the examination, the patient was also transfused with two units of fresh frozen plasma. Antibiotic therapy was normally continued, whereas multivitamin supplementation was terminated. In further coagulation studies, INR and PT were eventually reverted to baseline normal values.

In the meantime, severe diarrhea and abdominal tenderness were also observed, while stool testing for *Clostridium difficile* antigen and toxins was positive. Abdominal computed tomography revealed profound thickening and edema throughout the entire colonic wall, pneumatosis intestinalis, and ascites. With a diagnosis of pseudomembranous colitis, oral vancomycin was added to the antibiotic therapy, while the patient was transferred to the Department of General Surgery, where a permanent gastrostomy was performed after supportive treatment and recovery of the patient. He was eventually discharged, with instructions to complete the antibiotic regimen with vancomycin and rifaximin per gastrostomy.

## 2. Vitamin K and Its Function

Vitamins are a group of biochemically distinct families of structurally and functionally related organic compounds that are metabolically necessary in trace amounts. Along with essential amino acids, essential fatty acids, and minerals, they constitute the essential nutrients, as they cannot be synthesised in sufficient quantities by the organism per se but rather must be intaken by diet or obtained by symbiosis with the human microbiome [1].

Vitamin K is a family of two major vitamers, i.e., phylloquinone (vitamin K_1_) that is mainly contained in plants, especially green leafy vegetables, placed on the photosynthetic membranes of chloroplasts in plant cells, and menaquinone (vitamin K_2_) that is predominantly produced by specific bacterial species of the intestinal microflora, while it is also found in quite lower amounts in animal-derived dietary products. Menaquinone has multiple subtypes characterised as MK-4 through MK-13 based on the length of their isoprenoid side chain. The vast majority of facultative or obligate anaerobes in the gut, such as *Escherichia coli*, *Bacteroides fragilis*, *Eggerthella lenta*, and *Lactococcus lactis*, produce all different subtypes of menaquinones, whereas their imbalance seen in intestinal microflora dysbiosis can negatively impact the production [2,3].

As a class of lipid-soluble compounds, intact hepatobiliary, pancreatic, and intestinal function is a prerequisite for its unimpeded absorption (Figure 2). The ingested form of vitamin K_1_ is bound to proteins, which are lysed by pancreatic enzymes in the small intestine. Both free dietary vitamin K_1_ and bacteria-derived vitamin K_2_ are solubilised into micelles by bile salts and then are absorbed by enterocytes and incorporated into chylomicrons. It is transferred to the systemic circulation by lymphatic vessels, bypassing the portal circulation, and eventually transported to the liver, where it is incorporated into very low-density lipoprotein (VLDL) and in this form it re-enters the systemic circulation to be distributed in the body [2].

Vitamin K holds a central role in coagulation homeostasis. The hydroquinone form of vitamin K acts as a cofactor of the enzyme *gamma-glutamyl* carboxylase for the γ-carboxylation of glutamic acid residues on several proteins involved in the coagulation cascade, including factors II, VII, IX, and X, as well as proteins C, S, and Z, thus transforming them into their active forms (Figure 3). Molecular oxygen and carbon dioxide are also needed in this reaction, while one molecule of the epoxide form of vitamin K is produced per each glutamic acid residue that becomes carboxylated. The enzyme *vitamin K epoxide reductase* is responsible for the reduction of vitamin K-2,3-epoxide and the production of vitamin K hydroquinone. Consequently, vitamin K deficiency is mainly associated with hypocoagulation and increased haemorrhagic diathesis, and it also constitutes a therapeutic target for medically-induced anticoagulation, as it is in the case of warfarin and acenocumarol, among other vitamin K antagonists [4,5].

Vitamin K depletion in an otherwise healthy individual is rare, as phylloquinone is widely available in vegetal dietary products, menaquinone is produced by the intestinal microflora, and vitamin K is recycled within cells. However, hypovitaminosis K is common in newborns, so that administration of vitamin K prophylaxis at birth is necessitated to avoid it. Major causes of vitamin K deficiency include decreased dietary intake, decreased absorption by the gastrointestinal tract, hepatic dysfunction, and interactions with medications. Prolonged fasting or starvation, abused alcohol consumption, and prolonged parenteral nutrition without vitamin K supplementation can all contribute to decreased vitamin K intake. Malabsorptive states can lead to impaired vitamin K absorption and can be attributed to impaired bile or pancreatic secretion, such as in cystic fibrosis, primary biliary cholangitis, primary sclerosing cholangitis, and biliary atresia, as well as to intestinal diseases, such as celiac disease, inflammatory bowel disease, and short bowel syndrome, especially in case of terminal ileum involvement. Vitamin K depletion is also seen in chronic liver disease, while decreased hepatic production of coagulation factors can additionally lead to coagulation impairment in this condition. Finally, antibiotics and high vitamin E doses can contribute to hypovitaminosis K, as described below [2].

## 3. Antibiotic Therapy and Its Impact on Vitamin K Bioavailability and Function

In the first years after the introduction of the antibiotics in daily clinical practice, the use of broad-spectrum antibiotic therapy for prolonged time periods was observed to be associated with vitamin K deficiency. Since then, a considerable number of case reports and clinical studies have described the phenomenon. Vitamin K depletion was seen as a common complication in adult patients hospitalised in intensive care units (ICU) with an incidence rate that reached 25% [6]. Also, low vitamin K levels were found in hospitalised children of up to 12 months of age, who received prolonged antibiotic therapy [7].

At least two distinct underlying mechanisms of the interference of antibiotics with vitamin K bioavailability and function have been unravelled. Firstly, prolonged use of broad-spectrum antibiotics or combinations of antibiotics potentially lead to decreased vitamin K production and absorption by inducing intestinal microflora dysbiosis and eventually causing hypovitaminosis K. Secondly, antibiotic agents, particularly of the cephalosporin class, with 1-N-methyl-5-thiotetrazole (NMTT) or 2-methyl-1,3,4-thiadiazole (MTD) side groups, are capable of inhibiting vitamin K function (Figure 4).

### 3.1. Broad-Spectrum Antibiotic Therapy Potentially Leads to Intestinal Microflora Dysbiosis and Restriction of Vitamin K-Producing Bacterial Populations

In the literature, several clinical cases of vitamin K deficiency with or without resultant hypocoagulation attributed to long-term and/or wide spectrum antibiotic therapy are reported. Haden observed general haemorrhagic diathesis and haematochezia as well as hypoprothrombinemia that was corrected by vitamin K administration in a patient, who was operated for right ureteral obstruction and received multiple antibiotics postoperatively [8]. An unexpected vitamin K depletion was described in operated patients with poor food intake receiving antibiotics in the postoperative state [9]. Itagaki and Hagino published a case of a patient with haemorrhagic shock after thoracentesis, which was attributed to anatomic abnormality of the intercostal artery and hypovitaminosis K due to restricted diet and antibiotic therapy [10]. Vitamin K deficiency-associated hypocoagulation in a patient with extensive burns receiving meropenem was only reversed upon discontinuation of the broad-spectrum antibiotic therapy [11]. Nomoto et al. reported a case of a patient with schizophrenia and catatonia who was hospitalised due to aspiration pneumonia and demonstrated low vitamin K levels and INR prolongation resulting from antibiotic therapy with ampicillin/sulbactam and cefmetazole as well as long-term restriction of food intake due to negativism and stupor [12]. Furthermore, Quinn et al. showed in their study involving *Helicobater pylori*-infected mice that the antibiotic eradication of vitamin K producers in the intestinal tract, combined with a vitamin K-deprived diet, led to remarkable vitamin K depletion and severe gastrorrhagia in the animals [13].

### 3.2. Antibiotics with NMTT or MTD Side Groups Inhibit Vitamin K Function

It is known that older antibiotics, particularly of the cephalosporin class, which contain in their molecule the NMTT side group, such as cefamandole, cefmenoxime, cefoperazone, cefotetan, and moxalactam, or the MTD side group, such as cefazolin, inhibit the γ-carboxylation and activation of vitamin K-dependent clotting factors, thus leading to hypoprothrombinemia and increasing haemorrhagic risk [14,15,16]. Several studies highlighted the impact of hypovitaminosis K and NMTT-containing antibiotic treatment on coagulation status in rats. The administration of moxalactam to the animals fed with a vitamin K-deprived diet further enhanced the hypoprothrombinemic effects of vitamin K depletion, as shown by coagulation parameters and multiple organ haemorrhage [17]. The same antibiotic was also found to depress the liver microsomal vitamin K-2,3-epoxide activity, thus further contributing to hypoprothrombinemia under vitamin K-deficient conditions [18].

### 3.3. Prophylactic Administration of Vitamin K in Patients Receiving Antibiotic Therapy

Two Indian studies in paediatric populations previously investigated the utilisation of vitamin K supplementation for the prevention of hypocoagulation in patients receiving antibiotic therapy. In one study of 120 children aged between 2 months and 12 years, a single dose of vitamin K was proven ineffective in preventing vitamin K deficiency-induced hypoprothrombinemia [19]. Another study involving 80 neonates with sepsis, who were hospitalised in the ICU and received antibiotic therapy for at least 7 days, showed that they all demonstrated hypovitaminosis K that was resistant to intramuscular administration of vitamin K [20]. It was thus shown in both studies that vitamin K supplementation was incapable of preventing vitamin K deficiency and related coagulopathy in critically ill patients. On the other hand, in a review of Egyptian studies investigating intracranial haemorrhage associated with vitamin K depletion in infants, the prophylactic use of vitamin K in mothers at imminent risk of preterm labour as well as of a vitamin K boost in exclusively breastfed infants receiving prolonged antibiotic therapy or presenting persistent diarrhea were highlighted as potentially useful interventions for improving outcomes [21].

## 4. Vitamin E Supplementation and Its Impact on Vitamin K Bioavailability and Function

Interactions between vitamins E and K were firstly reported in 1945, when Woolley et al. observed that pregnant mice demonstrated reproductive tract haemorrhage and resorption of the fetus after administration of dl-α-tocopheryl quinone, a vitamin E derivative, a complication that was eventually prevented by co-administering vitamin K [22]. Since then, several animal studies and case reports have thoroughly described the interference of vitamin K function by vitamin E supplementation and the resultant devastating effects on coagulation homeostasis (Figure 4).

In a study involving one-day chicks, a diet with adequate amounts of vitamin K but high amounts of vitamin E for one month resulted in a threefold increase of PT, haemorrhagic tendency, and higher mortality rate, while coagulopathy was reversed upon administration of phylloquinone [23]. In another study series involving rats, dogs, and humans, vitamin E supplementation was capable of heightening hypoprothrombinemia in vitamin K-deficient animals or individuals, whereas it had no impact on the coagulation status of healthy subjects with sufficient levels of vitamin K [24,25].

It was earlier found that high doses of vitamin E enhance the actions of coumarin-based oral anticoagulants, such as warfarin, thus increasing haemorrhagic risk [26]. In a patient treated with warfarin for myocardial infarction and pulmonary embolism, a haemorrhagic syndrome including diffuse ecchymoses and haematuria was attributed to a vitamin E megadose and its interference with vitamin K [27]. Interestingly, coagulopathy and haemorrhage in a young patient taking non-steroidal anti-inflammatory drugs and a vitamin E supplement occurred due to low levels of vitamin K, however with only slightly high levels of vitamin E, indicating that even mild hypervitaminosis E could cause clinically significant inhibition of vitamin K-dependent coagulation factors and ultimately increase haemorrhagic risk, especially in the presence of other drugs with a similar side effect [28].

Traber proposed two main hypothetical mechanisms by which higher levels of vitamin E lead to lower levels of active vitamin K and, therefore, result in hypoprothrombinemia and haemorrhagic diathesis: (1) vitamin E interferes with the transformation of phylloquinone to menaquinone-4, either (a) by competing for the K_1_ side chain truncation by an undiscovered enzyme and/or (b) by competing for the K_1_ side chain ω-hydroxylation by a hypothetical cytochrome P450 enzyme thereby preventing its β-oxidation and removal; (2) vitamin E enhances hepatic metabolism and excretion of all vitamin K forms through xenobiotic pathways [29]. Also, interestingly, in a study involving rats on a high-vitamin E diet, the interaction between the two vitamins appeared to be tissue-dependent, as vitamin K levels were significantly lowered in the spleen, the kidneys, and the brain, but not in the plasma or the liver [30].

## 5. Conclusions and Future Perspectives

The association of hypovitaminosis K and resultant derangement of coagulation homeostasis with either broad-spectrum antibiotic therapy or vitamin E supplementation is highlighted in several studies on animals and humans as well as in case reports. There are several aspects of this phenomenon, however, that are yet to be unravelled, such as the precise mechanisms of vitamins E and K interference or the particular antibiotics that are more likely to induce intestinal microflora dysbiosis.

We report a malnourished patient with aspiration pneumonia treated with wide-spectrum antibiotic therapy and multivitamin supplementation, including vitamin E, who gradually developed severe INR prolongation that was rapidly reversed after the administration of vitamin K. This is the first time, to the best of our knowledge, that all iatrogenic risk factors for severe vitamin K deficiency-induced coagulopathy, except for the administration of vitamin K antagonists, are collectively found in a single clinical case. Furthermore, by sharing common features with the case described by Nomoto et al., it thus highlights the increased risk of coagulopathy associated with vitamin K depletion in malnourished patients with mental diseases who receive antibiotic therapy for long time periods.

Based on the available scientific evidence, we recommend that close monitoring of coagulation parameters, i.e., INR, PT, and APTT, be performed in patients who receive wide-spectrum antibiotic therapy, especially in those who have underlying malnutritive or malabsorptive conditions. We further recommend that antibiotics with NMTT or MDT side groups are avoided in those patients, and vitamin E supplementation is only administered if absolutely indicated, considering its potential interference with vitamin K bioavailability and function. Declining haematocrit and haemoglobin levels and possible clinical signs of haemorrhage should be assessed as indirect findings of coagulopathy. In case of a noticeable coagulation impairment, possible discontinuation of antibiotic therapy and switching to another regimen should be considered on a case-by-case basis. Therapeutic administration of vitamin K would be appropriate as soon as hypovitaminosis K is confirmed or highly suspected. Further studies are needed to define the antibiotics that are more likely to induce a vitamin K deficiency-associated coagulopathy, the proper time intervals for the monitoring of coagulation parameters, and the possible beneficial use of prophylactic supplementation with vitamin K, specifying the proper dosage and route of administration.

## Figures and Tables

**Figure 1 jpm-13-01349-f001:**
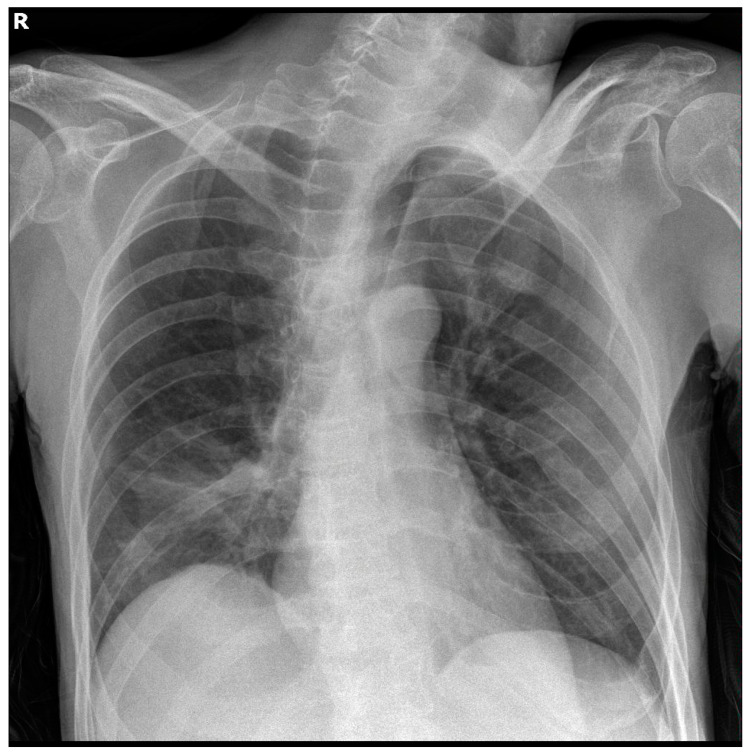
Chest imaging of the patient. Chest X-ray showed opacities in the right lower lung field, a finding compatible with aspiration pneumonia.

**Figure 2 jpm-13-01349-f002:**
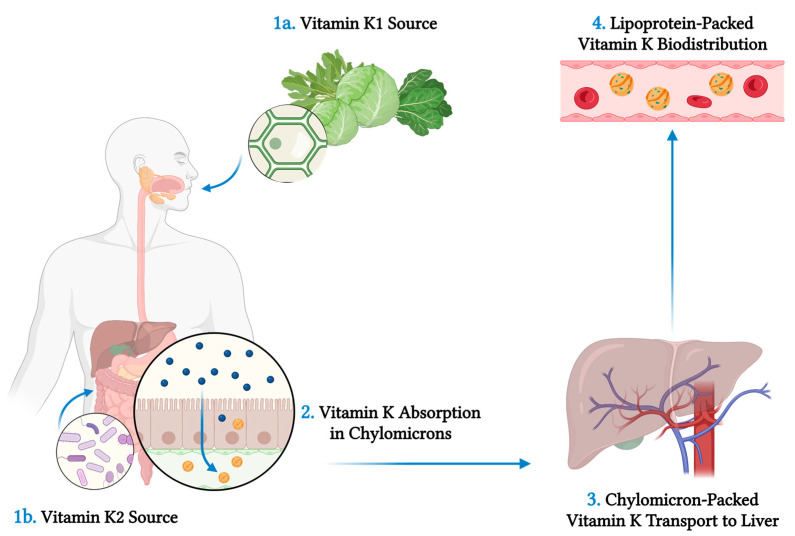
Vitamin K sources and metabolism. Vitamin K_1_ (phylloquinone) is intaken by consumption of plants, especially green leafy vegetables, as it is found on the photosynthetic membranes of chloroplasts in plant cells (**1a**). Vitamin K_2_ (menaquinone) is produced by specific bacterial species in the intestinal microflora (**1b**). Vitamin K_1_ is bound to proteins, which are lysed by the action of pancreatic enzymes in the small intestine. Both dietary and bacteria-derived vitamin K is absorbed by enterocytes after it is solubilised into micelles by bile salts (**2**). Vitamin K is then incorporated into chylomicrons, transferred to the systemic circulation through lymphatic vessels, bypassing the portal circulation, and eventually transported to the liver (**3**). In the liver, vitamin K is Incorporated into very low-density lipoproteins (VLDL) and in this form it re-enters the systemic circulation and it is distributed to the body (**4**). Created with: BioRender.com (accessed on 30 August 2023).

**Figure 3 jpm-13-01349-f003:**
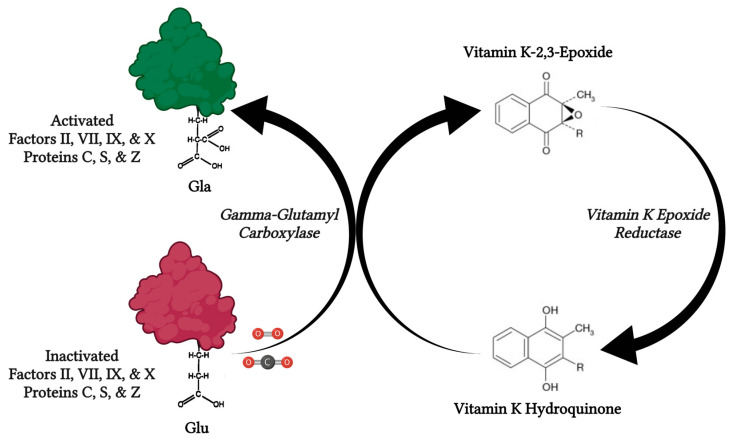
The role of vitamin K in the coagulation pathway. Vitamin K-dependent coagulation factors II, VII, IX, and X as well as proteins C, S, and Z are transformed into their active forms by the carboxylation of the glutamic acid residues in their molecules. In this reaction, which is catalysed by vitamin K-dependent *gamma-glutamyl carboxylase*, molecular oxygen (O_2_) and carbon dioxide (CO_2_) also take part, while vitamin K is oxidised from vitamin K hydroquinone into vitamin K-2,3-epoxide. The levels of the active form of vitamin K are replenished by the action of *vitamin K epoxide reductase*, which reduces vitamin K-2,3-epoxide into vitamin K hydroquinone. Created with: BioRender.com (accessed on 30 August 2023).

**Figure 4 jpm-13-01349-f004:**
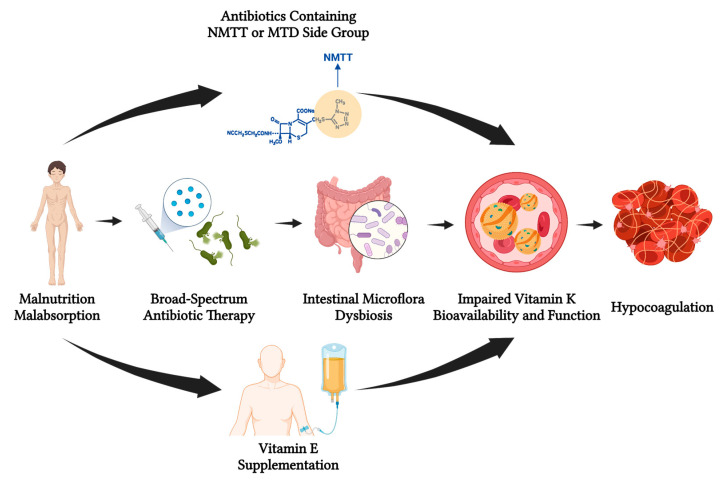
Factors leading to low vitamin K bioavailability and interfering with its function. Malnutritive or malabsorptive states result in decreased dietary intake of vitamin K (mostly vitamin K_1_) or intestinal absorption of vitamin K (both vitamins K_1_ and K_2_), respectively. The use of broad-spectrum antibiotic therapy potentially leads to dysbiosis and restriction of the vitamin K_2_ producers in intestinal microflora. Antibiotics containing the NMTT or MTD side groups in their molecule additionally inhibit the γ-carboxylation and activation of vitamin K-dependent clotting factors. Furthermore, vitamin E supplementation in high doses unfavourably interacts with vitamin K, yet with undetermined mechanisms, while vitamin K antagonists, such as warfarin among other anticoagulants, inhibit the conversion of vitamin K in its active reduced form (not shown in this figure). Low vitamin K bioavailability and impairment of its function eventually lead to disturbance of coagulation homeostasis and heightening of haemorrhagic risk. Abbreviations: NMTT: 1-N-methyl-5-thiotetrazole; MTD: 2-methyl-1,3,4-thiadiazole. Created with: BioRender.com (accessed on 30 August 2023).

**Table 1 jpm-13-01349-t001:** Laboratory examinations of the patient on selected days of his hospitalisation.

Parameters	Units	1st Day	3rd Day	13th Day	19th Day
Complete Blood Count					
Haemoglobin	g/dL	14.30	11.80	12.90	11.40
Haematocrit	%	43.20	37.30	40.80	36.80
RBC	C/mm^3^	5.06 × 10^6^	4.25 × 10^6^	4.65 × 10^6^	4.14 × 10^6^
PLT	C/mm^3^	193 × 10^3^	189 × 10^3^	832 × 10^3^	382 × 10^3^
WBC	C/mm^3^	12.88 × 10^3^	13.89×10^3^	17.50 × 10^3^	8.15 × 10^3^
Neutrophils	C/mm^3^	9.41 × 10^3^	11.16 × 10^3^	15.10 × 10^3^	6.26 × 10^3^
Lymphocytes	C/mm^3^	1.76 × 10^3^	1.08 × 10^3^	1.40 × 10^3^	1.18 × 10^3^
Monocytes	C/mm^3^	1.54 × 10^3^	1.53×10^3^	0.83 × 10^3^	0.56 × 10^3^
Eosinophils	C/mm^3^	0	0.01 × 10^3^	0.03 × 10^3^	0.03 × 10^3^
Basophils	C/mm^3^	0.03 × 10^3^	0.02 × 10^3^	0.03 × 10^3^	0.02 × 10^3^
Biochemical Studies					
Glucose	mg/dL	88.00	95.00	65.00	183.00
Albumin	g/dL	-	3.50	3.80	2.60
Total Proteins	g/dL	-	6.20	-	4.60
Urea	mg/dL	23.00	33.00	14.00	98.00
Creatinine	mg/dL	0.60	0.50	0.40	1.20
Sodium	mmol/L	134.00	138.00	136.00	148.00
Potassium	mmol/L	3.90	3.90	4.40	4.60
SGOT	IU/L	23.00	27.00	66.00	67.00
SGPT	IU/L	22.00	24.00	48.00	48.00
ALP	IU/L	-	82.00	-	-
GGT	IU/L	54.00	76.00	84.00	128.00
Total Bilirubin	mg/dL	-	0.60	0.40	0.50
Amylase	IU/L	-	142.00	185.00	-
LDH	IU/L	161.00	150.00	403.00	201.00
CPK	IU/L	-	146.00	566.00	152.00
CRP	mg/dL	11.69	39.37	4.39	27.41
PCT	ng/mL	0.09	1.87	-	-
Coagulation Studies *					
INR	-	1.35	-	4.17	7.94/2.86
PT	s	17.50	-	50.80	91.60/35.20
APTT	s	57.50	-	52.50	66.20/69.30

* On the 19th day, coagulation studies before and after administration of vitamin K are shown. Abbreviations: ALP: alkaline phosphatase; APTT: activated partial thromboplastin time; CPK: creatinine phosphokinase; CRP: C-reactive protein; GGT: gamma-glutamyl transferase; INR: international normalised ratio; LDH: lactate dehydrogenase; PCT: procalcitonin; PLT: platelets; PT: prothrombin time; RBC: red blood cells; SGOT: serum glutamic oxaloacetic transaminase; SGPT: serum glutamic-pyruvic transaminase; WBC: white blood cells.

## Data Availability

No new data were created or analyzed in this study. Data sharing is not applicable to this article.

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
