# Peer review of "Association of Broad-Spectrum Antibiotic Therapy and Vitamin E Supplementation with Vitamin K Deficiency-Induced Coagulopathy: A Case Report and Narrative Review of the Literature"

_jpm, 2023, doi:10.3390/jpm13091349_

Round 1

Reviewer 1 Report

Dear Authors,

Thank you for your manuscript. The following are my input:

1. Case Report: is there any result to showed the subsequent normalisation of the result of the PT? Was it reverted to baseline after completion of antibiotic? was the Vitamin E still continued? Was the patient under-nourished being incapacitated?

2. Section 2: Page 4: Line 103-109- this is a mistake of copying the materials and methods instruction.

3. Some minor English corrections: please look at the manuscript for some mistake, eg: Section 3, Page 5, Line 140, the first Early word-probably is not rightly place.

4. Figure 3: the error for antibiotics containing NMTT should originate from the broad spectrum antibiotic icon instead of malnutrition icon.

5. For section 3.1: probably can list down some evidence to suggest dysbiosis in the cases of patients with this antibiotic treatment? Also can list down organism that are important in vitamin K function-and show the impact of the antibiotic treatment towards this particular organism. Was it also contribute to his underlying nutritional state that predispose to the dysbiosis before the antibiotic?

6. For section 3.3: please conclude findings of these two studies-as their findings were not in the same direction with your treatment outcome.

A good case to review the literature and creating awareness. All the best

As mentioned, only minor involving several corrections need to be look at, mainly accidental inclusion of probably wrong words. 

Author Response

  1. We thank the Reviewer for their comment. Further serial assessments of coagulation status revealed restoration of INR and PT to their baseline normal values. Antibiotic therapy was normally continued, whereas multivitamin supplementation was terminated. The patient was malnourished (as reported in lines 42 and 244) due to poor food intake attributed to schizophrenia and catatonia. This information has been added to the manuscript (in highlight).
  2. We appreciate the Reviewer’s comment. Indeed, there was a mistake of copying. We have revised the text accordingly.
  3. We thank the Reviewer for their comment. The manuscript has been rechecked for minor typos, which have been corrected accordingly (in highlight).
  4. We appreciate the Reviewer’s comment. In figure 3, we placed the arrow for NMTT-containing antibiotics to originate from the malnutrition/malabsorption icon and not the broad-spectrum antibiotic therapy icon, as NMTT-antibiotics are not necessarily broad-spectrum antibiotics and could interfere with vitamin K function in malnourished patients irrespective to intestinal microflora dysbiosis, as explained in figure 3 legend and 3.3 section.
  5. We thank the Reviewer for their comment. Some examples of vitamin K-producing bacteria in intestinal microflora have been added in section 2 (in highlight). As these include the vast majority of facultative and obligate anaerobes, and the patient received multiple combinations of broad-spectrum antibiotics, as described in section 1, it would be difficult to clearly delineate the precise effect of the therapy on each microorganism. Indeed, most probably the underlying malnutrition condition of the patient contributed to the vitamin K deficiency-induced coagulopathy. This is mentioned in the Conclusions and Future Perspectives section.
  6. We appreciate the Reviewer's comment. A conclusion sentence has been added at the end of the paragraph (in highlight).

Reviewer 2 Report

This is only a case report, I don't know what value it can have in terms of scientific literature

Vitamin K deficiency is a problem in critically ill patients

You could enrich the case report with a case series review

How does Vitamin E interfere with Vitamin K deficiency?

Author Response

We thank the Reviewer for their comment. Additional cases to be included in a case series would be, indeed, very useful to enrich our article. However, we only have this single case to report at this moment. The unique feature of this case though is that both iatrogenic causes of vitamin K deficiency, i.e. broad-spectrum antibiotic therapy and vitamin E supplementation, contributed to coagulopathy in the patient, whereas other clinical cases reported in the literature describe either of the two risk factors (not both) as a cause per single case. The exact mechanisms of vitamins E and K interference is not known. The two major hypotheses, as described in the literature, are reported in the last paragraph of section 4.

Reviewer 3 Report

1. The study is a case report and narrative review for association between antibiotic therapy and vitamin emE supplementation   and coagulopathy. However, the major concern in this study is the affected type of vitamin K. There are two forms of Vitamin K. Vitamin K1 and called phylliquinone and Vitamin K2 called menaquinones. It was not clear which type given to the case. The role of all vitamin K forms has to be clarified.

2. It was necessary to analyze endogenous vitamin K for the case before and after the treatment. Levels of PK, MK-4 and MK-7 is necessary to clarify their roles.

3. A figure show the possible roles of each vitamin K form in the study.

Author Response

  1. We thank the Reviewer for their comment. The patient was intravenously administered with vitamin K1, as reported in section 1. The different forms, sources, and metabolism of vitamins K1 and K2 have been explained in section 2 (in highlight).
  2. We appreciate the Reviewer’s comment. It would be indeed useful to analyse endogenous vitamin K, including PK, MK-4 and MK-7, before and after treatment with vitamin K. This was not performed due to lack of reagent in our laboratory. We rather used the measurement of vitamin K-dependent coagulation factor VII to indirectly assess vitamin K deficiency. This was, of course, confirmed with the correction of INR after the administration of vitamin K as our sole intervention (before the administration of fresh frozen plasma).
  3. We thank the Reviewer for their comment. A figure showing the different types and sources of vitamin K has been included (figure 2). Details have also been added on section 2 (in highlight).

Round 2

Reviewer 3 Report

The required changes were made and the manuscript could be accepted in its present form